# WHAT SHAPES THE LOSS LANDSCAPE OF SELF SUPERVISED LEARNING?

Liu Ziyin[1,2,3†], Ekdeep Singh Lubana[2,3,4†], Masahito Ueda[1,5,6], Hidenori Tanaka[2,3]

[1]*Department of Physics, The University of Tokyo, Tokyo, Japan*
[2]*Physics & Informatics Laboratories, NTT Research, Inc., Sunnyvale, CA, USA*
[3]*Center for Brain Science, Harvard University, Cambridge, USA*
[4]*EECS Department, University of Michigan, Ann Arbor, USA*
[5]*Institute for Physics of Intelligence, The University of Tokyo, 7-3-1 Hongo, Bunkyo-ku, Tokyo*
[6]*RIKEN Center for Emergent Matter Science (CEMS), Wako, Saitama, Japan*

## ABSTRACT

Prevention of complete and dimensional collapse of representations has recently become a design principle for self-supervised learning (SSL). However, questions remain in our theoretical understanding: When do those collapses occur? What are the mechanisms and causes? We answer these questions by deriving and thoroughly analyzing an analytically tractable theory of SSL loss landscapes. In this theory, we identify the causes of the dimensional collapse and study the effect of normalization and bias. Finally, we leverage the interpretability afforded by the analytical theory to understand how dimensional collapse can be beneficial and what affects the robustness of SSL against data imbalance.

## 1 INTRODUCTION

Self-supervised learning (SSL) methods have achieved remarkable success in learning good representations without labeled data (Chen et al., 2020b). Loss functions used in such SSL techniques promote representational similarity between pairs of related samples while using explicit penalties (Chen et al., 2020a; He et al., 2020; Zbontar et al., 2021; Caron et al., 2020) or asymmetric dynamics (Caron et al., 2021; Grill et al., 2020; Chen and He, 2021) to ensure that the distance between unrelated samples remains large. In practice, however, SSL training often experiences the phenomenon of *dimensional collapse* (Jing et al., 2021; Tian et al., 2021; Pokle et al., 2022), where the learned representation spans a low dimensional subspace of the overall available space. In the extreme case, this failure mode instantiates as a *complete collapse*, where the learned representation becomes zero-rank, and no informative features can be extracted.

Prior work has primarily positioned such collapses in SSL as enemies of learning, arguing that they can negatively impact downstream task performance (Zbontar et al., 2021; Jing et al., 2021; Bardes et al., 2021). However, recent work by Cosentino et al. (2022) empirically demonstrates otherwise: quality of representations can be improved when there is a degree of collapse. These conflicting results indicate that despite extensive empirical explorations, a gap remains in our understanding of the collapse phenomenon in SSL training. We argue that this gap is due to the lack of a theoretical framework to analyze the mechanisms promoting collapsed representations. We aim to close this gap by carefully studying the loss landscapes of SSL.

In this work, we analytically solve the effective landscapes of linear models trained on several popular losses used in self-supervised learning, including InfoNCE (Oord et al., 2018), Normalized Temperature Cross-Entropy (NT-xent) (Chen et al., 2020a), Spectral Contrastive Loss (HaoChen et al., 2021), and Barlow Twins / VICReg (Zbontar et al., 2021; Bardes et al., 2021). The main thesis of this work is: *the local geometry of the SSL landscapes around the origin crucially decides the learning behavior of SSL models*. Technically, we show that

1. the interplay between data variation and data augmentation determines the geometry of the loss;
2. the geometry of the loss explains when dimensional collapse can be helpful and why certain SSL losses are robust against data imbalance, but not the others.

To the best of our knowledge, our work is the first to study the landscape causes of collapse in SSL thoroughly.

†Work done during an internship at Physics & Informatics Laboratories, NTT Research.

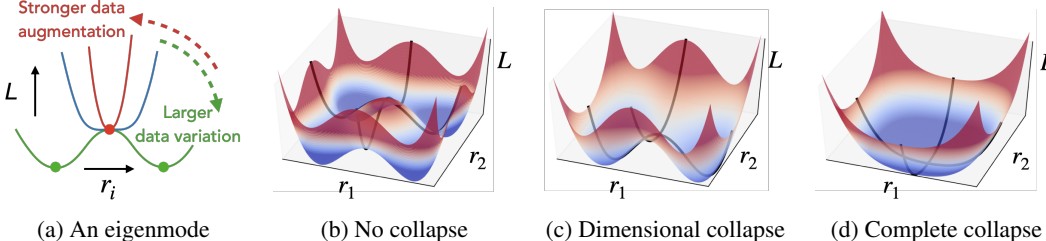

(a) An eigenmode      (b) No collapse      (c) Dimensional collapse      (d) Complete collapse

Figure 1: **Landscape in self-supervised learning (SSL).** SSL losses generally depend only on the relative angle between pairs of network outputs (e.g, $f(x)^T f(x')$). Thus, the landscapes with a linear network ($f(x) = Wx$) have a global rotational symmetry and are symmetric about the origin. Our theory finds that the local stability at the origin decides the collapse, and larger data variation (green) prevents collapse, while strong data augmentation (red) can promote collapse. We plot the loss for a toy linear model with a diagonal weight matrix $diag(r_1, r_2)$. (a) The $1d$ landscape when fixing one of the parameter. (b-d) The $2d$ landscape. (b) No collapse: the origin is an unstable local maximum, and surrounding local minima avoid collapse. The dimensionally collapsed solutions are the saddle points. (c) Dimensional collapse: the value of $w_1$ for all stable fixed points collapses to zero. (d) Complete collapse: the origin becomes the isolated local minimum.

## 2    RELATED WORKS

**SSL and Collapses.** On the one hand, prior literature has often argued collapse as a harmful phenomenon that can deteriorate downstream task performance (Jing et al., 2021; Zbontar et al., 2021). Preventing such collapsed representations is a frequently discussed topic in literature (Hua et al., 2021; Jing et al., 2021; Pokle et al., 2022; Tian et al., 2021) and has motivated the design of several SSL techniques (Zbontar et al., 2021; Bardes et al., 2021; Ermolov et al., 2021). On the other hand, Cosentino et al. (2022) empirically showed that dimensional collapses under strong augmentations could significantly improve generalization performance. Our work demystifies these conflicting results by finding analytic solutions to loss landscapes of several standard SSL techniques.

**Theoretical Advances in SSL.** Recently, several advances have been made towards understanding the success of SSL techniques from different perspectives: e.g., learning theory (Arora et al., 2019; Saunshi et al., 2022; Nozawa and Sato, 2021; Wei et al., 2021), information theory (Tsai et al., 2021a;b; Tosh et al., 2021), causality and data-generating processes (Zimmerman et al., 2021; Kugelgen et al., 2021; Trivedi et al., 2022; Tian et al., 2020; Mitrovic et al., 2020; Wang et al., 2022), dynamics (Wang and Isola, 2020; Tian et al., 2021; Tian, 2022; Wang and Liu, 2021; Simon et al., 2023), and loss landscapes (Pokle et al., 2022). These advances have unveiled practically useful properties of SSL, such as robustness to dataset imbalance (Liu et al., 2021) and principled solutions to avoid spurious correlations (Robinson et al., 2021).

The work by Jing et al. (2021) is the closest to ours in problem setting. In that paper, the authors focused on studying the linearized learning dynamics and suggested that a competition between the feature signal strength and augmentation strength can lead to dimensional collapse. In contrast, our focus is on the landscape and our result implies that this feature-augmentation competition on its own is insufficient to cause a dimensional collapse. In fact, we show that there will be no collapse in the setting studied by Jing et al. (2021).

## 3    A LANDSCAPE THEORY OF SELF-SUPERVISED-LEARNING

This section presents the main theoretical results. Let $\{\hat{x}_i\}_i^N$ be a dataset with $N$ data points. For every data point $\hat{x}$, we augment it with an i.i.d. noise $\epsilon$ such that $x := \hat{x} + \epsilon$. To be concrete, we start with considering the standard contrastive loss, InfoNCE (Oord et al., 2018):

$$L = \mathbb{E}_\epsilon \left[ -\sum_{i=1}^N \log \frac{\exp(-|f(x_i) - f(x'_i)|^2/2)}{\sum_{j \neq i} \exp(-|f(x_i) - f(\chi_j)|^2/2) + \exp(-|f(x_i) - f(x'_i)|^2/2)} \right], \quad (1)$$

where $f(x) \in \mathbb{R}^{d_1}$ is the model output; all $x$, $x'$ and $\chi$ are augmented data points for some independent additive noise $\epsilon$ such that $\mathbb{E}_\epsilon[x] = \hat{x} = \mathbb{E}_\epsilon[x'] \neq \mathbb{E}_\epsilon[\chi] = \hat{\chi}$. We decompose the model output into a general function $\phi(x) \in \mathbb{R}^{d_0}$ and the last-layer weight matrix $W \in \mathbb{R}^{d_1 \times d_0}$: $f(x) = W\phi(x)$. The covariance of $\phi(\hat{x})$ is $A_0 := \mathbb{E}_{\hat{x}}[\phi(\hat{x})\phi(\hat{x})^T]$, and the covariance of the data-augmented penultimate layer representation is $\Sigma := \mathbb{E}_x[\phi(x)\phi(x)^T]$. The effect of data augmentation on the learned

representation is captured through a symmetric matrix $C := \Sigma - A_0$. For a general $\phi$, the eigenvalues of $C$ can be either positive or negative. When $\phi$ is the identity mapping, $A_0$ becomes the empirical data covariance, $C$ becomes positive semi-definite and is the covariance of the noise $\epsilon$, and $\Sigma$ is the covariance of the augmented data. In some sense, this loss function captures the essence of SSL: the numerator encourages the representation $f(x)$ to be closer to the representation of similar data, and the denominator encourages a separation between dissimilar data.

For a fixed set of noises, we can write the InfoNCE in a cleaner form:

$$L_\epsilon = \mathbb{E}_{\hat{x}} \left\{ \frac{1}{2}|f(x) - f(x')|^2 + \log \mathbb{E}_{\hat{\chi}} \left[ \exp\left( -\frac{1}{2}|f(x) - f(\chi)|^2 \right) \right] \right\}, \quad (2)$$

where we used $\mathbb{E}_{\hat{x}}$ to denote an averaging over the training set.

In this notation, we have $\mathbb{E}_\epsilon \mathbb{E}_{\hat{x}}[x] = \mathbb{E}_x[x]$ and $\mathbb{E}_\epsilon[L_\epsilon] = L$. We first show that the expansion of the loss function around the origin takes a rather universal form. We then find analytical solutions to the stationary points of this landscape and study their relevance to feature learning and collapses. See Table 1 for a summary of the main results. The proofs are presented in Appendix E. For a quantitative understanding, we mainly focus on the case when $\phi$ is the identity function. We discuss the general nonlinear case in Section 4.1.

| | Hessian | Dim. | Complete |
|---|---|---|---|
| InfoNCE | $A_0$ | ✗ | ✗ |
| NT-Xent (SimCLR) | $A_0 - C/N$ | ✓ | ✓ |
| Spectral Contrastive | $C$ | ✗ | ✗ |
| Barlow Twins | $A_0 + C$ | ✗ | ✗ |
| + Normalization | - | ✓ | ✗ |
| + bias | - | ✓ | ✓ |
| + Weight Decay | $+\gamma I$ | ✓ | ✓ |

Table 1: **What shapes the SSL landscapes around the origin?** For each of the SSL losses, the combination of data covariance ($A_0$), data-augmentation covariance ($C$), and dataset size ($N$) can affect its stability and thus determine the presence (✓) and absence (✗) of dimensional/complete collapse (Here, a ✓means "there exists a hyperparameter setting and data distribution such that the relevant collapse happens;" see section 3). Beyond collapses, the theory implies that SCL, whose landscape is formed primarily by data augmentation, is more robust to data imbalance than InfoNCE, which is affected primarily by the data (see section 4).

### 3.1 LANDSCAPE OF A LINEAR MODEL

We first analyze representative SSL loss functions and show that to leading order in $W$, the local geometry of SSL losses takes the following form

$$L = -\text{Tr}[WBW^T] + \frac{1}{8}\text{Var}[|W(x-\chi)|^2]. \quad (3)$$

A distinctive feature of Eq. (3) is that its first and third-order terms vanish. This is because the loss function is invariant to a left rotation of $W$. We will see that this symmetry in rotation is a crucial and general feature of the SSL loss functions that allow us to treat them in a universal way. We discuss how rotation symmetry can cause collapses in nonlinear settings in Section 4.

**InfoNCE.** The loss function simplifies to:

$$L = \underbrace{\text{Tr}[WCW^T]}_{E} + \underbrace{\mathbb{E}_{\epsilon,\hat{x}} \left\{ \log \mathbb{E}_{\hat{\chi}} \left[ \exp\left( -\frac{1}{2}|W(x-\chi)|^2 \right) \right] \right\}}_{-S}. \quad (4)$$

Expanding the entropy term to the fourth order, we obtain[1]

$$-S = -\mathbb{E}_x \mathbb{E}_\chi \left[ \frac{1}{2}|W(x-\chi)|^2 \right] + \frac{1}{8}\text{Var}[|W(x-\chi)|^2] + O(\|W\|^6). \quad (5)$$

This (perturbative) decomposition of entropy deserves some special attention. The entropy decomposes into a repulsion term that is second order in $W$, and a variance term that is fourth order in $W$. The first term encourages a repulsion between $x$ and its augmentation, which counteracts the effect of the energy term. The repulsion term can be decomposed into

$$\mathbb{E}_x \mathbb{E}_\chi \left[ \frac{1}{2}|W(x-\chi)|^2 \right] = \text{Tr}[WCW^T] + \text{Tr}[WA_0W^T]. \quad (6)$$

The first term encourages an expansion of $W$ along the direction of the augmentation $C$, while the second term encourages an expansion along the directions of feature $A_0$. It is intriguing to see

---

[1]Throughout, we use $\|\cdot\|$ to denote the $L_2$ norm for vectors and Frobenius norm for matrices.

that the repulsion term dominates the attraction of the energy term: the motion along the direction of $C$ completely cancels out, and only the expansion along $A_0$ remains. This means that to leading order, the learned representation has a larger variation along the directions where the data has a larger variation, which is what one naively expects. Collecting results, we have obtained the loss landscape in the neighborhood of the origin as $L = -\text{Tr}[WA_0W^T] + \frac{1}{8}\text{Var}[|W(x - \chi)|^2] + O(\|W\|^6)$.

**NT-xent (SimCLR)**. As an additional example, we analyze Normalized Temperature Cross-Entropy loss (NT-xent) used in SimCLR (Chen et al., 2020a). Tian (2022) shows that InfoNCE can be generalized to encompass NT-xent as follows:

$$L = \mathbb{E}_\epsilon \left[ -\sum_{i=1}^N \log \frac{\exp(-|f(x_i) - f(x_i')|^2/2)}{\sum_{\chi \neq x} \exp(-|f(x_i) - f(\chi_j)|^2/2) + \alpha \exp(-|f(x_i) - f(x_i')|^2/2)} \right]. \quad (7)$$

In contrast to InfoNCE, here one of the terms in the denominator is reweighted by a factor of $\alpha \geq 0$. Two interesting limits are $\alpha = 1$, where we recover the InfoNCE loss, and $\alpha = 0$, where we obtain NT-xent. For general $\alpha$, we refer to this loss as the *weighted InfoNCE*. We will see in section 3 that this weighted InfoNCE can have a mild dimensional collapse problem.

The same perturbative expansion as Eq. (4)–(6) gives

$$L = \frac{1-\alpha}{N}\text{Tr}[WCW^T] - \text{Tr}[WA_0W^T] + \frac{1}{8}\text{Var}[|W(x-\chi)|^2] + O(\|W\|^6) + O(\|W\|^4 N^{-1}). \quad (8)$$

Now, the Hessian of the origin is no longer guaranteed to be negative definite. In fact, if $\frac{1-\alpha}{N}C - A_0 \geq 0$, $W = 0$ becomes an isolated local minimum.

**Landscape Analysis**. The above discussion shows that the commone loss landscapes in self-supervised contrastive learning can be reduced to an effective form in Eq. (3). The following proposition shows that the variance term of the loss takes a specific form when the data is Gaussian.

**Proposition 1.** *Let the data and noise be Gaussian. Then, $L = -\text{Tr}[WBW^T] + \text{Tr}[W\Sigma W^T W\Sigma W^T]$.*

When the training ends, one expects the model to locate at (at least close to) a stationary point of the loss. It is thus important to identify all the stationary points of this loss function.

**Theorem 1.** *Let $d^* := \min(d_0, d_1)$. Let the data and noise be Gaussian. All stationary points $W$ of Eq. (3) satisfy $W^TW = \frac{1}{2}\Sigma^{-1/2}UM\Lambda U^T\Sigma^{-1/2}$, where $U\Lambda U^T$ is the eigenvalue decomposition of $\Sigma^{-1/2}B\Sigma^{-1/2}$, and $M$ is an arbitrary (masking) diagonal matrix containing only zero or one such that (1) $M_{ii} = 0$ if $\Lambda_{ii} < 0$ and (2) contain at most $d^*$ nonzero terms.*

*Additionally, if $C$ and $A_0$ commute, all stationary points satisfy*

$$W^TW = \frac{1}{2}\Sigma^{-1}B_M\Sigma^{-1}, \quad (9)$$

*where $B_M$ denotes the matrix obtained by masking the eigenvalues of $B$ with $M$.*

This stationary-point condition implies the direct cause of the dimensional collapse. Namely, dimensional collapse happens when the eigenvalues of the matrix $B$ become negative. The eigenvalues of $B$, in turn, depend on the competition between data augmentation and the data feature. Comparing the commuting case with the noncommuting case, we see that the main difference is that when $C$ does not commute with $A_0$, the augmentation can also change the orientation of the learned representation; otherwise, augmentation only affects the eigenvalues. To focus on the most important terms, we now assume that the augmentation is well-aligned with the features such that the augmentation covariance commute with the data covariance.

**Assumption 1.** *From now on, we assume $CA_0 = A_0C$.*

For the case of weighted InfoNCE, we have that $B = A_0 - \frac{1-\alpha}{N}C$. Let $a_i$ denote the $i$-th eigenvalue of the $A$ and $c_i$ that of $C$ viewed in a predetermined order; then, the $i$th subspace collapses when $\frac{1-\alpha}{N}c_i \geq a_i$, namely, when the variation introduced by the noise dominates that of the original data. Importantly, this collapse is a property shared by *all* stationary points of the landscape, and one cannot hope to fix the problem by, say, biasing the gradient descent towards a certain type of local minima. When weight decay is used, the condition for collapse becomes $\frac{1-\alpha}{N}c_i + \gamma \geq a_i$: it becomes easier to cause a collapse when weight decay is used.

The global minimum of the loss function is also easy to find. For all stationary points, the loss function takes a simple form; $L = -\frac{1}{4}\text{Tr}[\Sigma^2 B_M B]$. Thus, $L$ becomes more and more negative if the eigenvalues of $B_M$ align with the largest eigenvalues of $B$. Namely, the global minimum is achieved if $M$ leaves the largest eigenvalues of $B$ intact.

Because the stationary points contain collapsed solutions where the eigenvalues of $W^T W$ are zero, one is naturally interested in how likely it is to converge to these solutions.

**Proposition 2.** ($W^T W$ achieves maximum possible rank) *Let $m$ denote the number of positive eigenvalues $B$. Then,* $\text{rank}(W^T W) = \min(m, d^*)$ *for any local minimum.*

This proposition implies that the loss landscape of contrastive SSL (with a linear model) is rather benign because all local minima must achieve a maximum possible rank. In fact, this result implies that the collapses may be well controllable by carefully controlling and tuning the eigenvalues of the matrix $B$, which directly depends on the nature of the data augmentation we use.

### 3.2 Landscape with Normalization

It is common in practice to normalize the learned representation such that $\|f(x)\|^2 = c$. When normalization is applied, only the direction of the learned representation matters. While this is a simple trick in practice, its implication on the landscape is poorly understood. In this section, we extend our theory to analyze the effect of normalization.

We model the effect of normalization as a regularization term: $R := (\mathbb{E}_x \|f(x)\|^2 - c)^2$:

$$L_{\text{norm}} = Eq.\ (3) + \kappa R. \tag{10}$$

Note that this regularization term achieves two things simultaneously: (1) $\|f(x)\|^2 = c$ for all $x$ is a minimizer of the loss function; (2) the regularization is invariant to any rotation of the learned representation. For a linear model, we note that this condition is not entirely the same as a direct normalization of the representation because it is generally impossible to achieve $\|Wx\|^2 = c$ for all $x$ because a linear model has limited expressivity. However, it is generally possible to achieve the slightly weaker condition: the representation has a norm 1 on average. This loss function can also be seen as a mathematical model of the VICReg loss (Bardes et al., 2021), where $R$ effectively models the variance regularization term of VICReg loss and $\kappa$ is its strength. This modeling is necessary because the variance term of the original VICReg is not differentiable and thus cannot be expanded. The proposed term $R$ captures the essence of the variance term because it also encourages the representation to have a constant variance. Our theory also explains why the VICReg is observed to experience collapses when $\kappa$ is not large enough. As $\kappa$ tends to infinity, this constraint will become perfectly satisfied. We thus take the infinite $\kappa$ limit to study the effect of normalization.

The following proposition gives a condition that all stationary points of Eq. (10) satisfy.

**Proposition 3.** *Let $\rho(W) := \text{Tr}[W\Sigma W^T]$, $B' := B + 2\kappa(c-\rho)\Sigma$, and let $\Lambda_i$ be the eigenvalues of $B'$. Then, every stationary point of Eq. (10) satisfy $W^T W = \frac{1}{2}\Sigma^{-1} B'_M \Sigma^{-1}$, where $M$ is an arbitrary diagonal mask of the eigenvalues of $B'$ containing only zero or one such that (1) $M_{ii} = 0$ if $\Lambda_i < 0$ and (2) contain at most $d^*$ nonzero terms.*

Compared with the unnormalized case, the term $2\kappa(1-\rho)\Sigma_M$ emerges due to normalization. The effect of normalization is as expected: it shrinks the norm of the model if $\rho > 1$, and it expands the model if $\rho < 1$, and it does not have any effect if we have already achieved $\rho = 1$. Interestingly, this rescaling effect is anisotropic and stronger along the directions of larger eigenvalues of the covariance of the augmented data $\Sigma$.

The next theorem gives the explicit form of $\rho$ at the stationary points.

**Proposition 4.** *For any stationary point $W^*$, $c - \rho(W^*) = \frac{c - \frac{1}{2}\text{Tr}[\Sigma^{-1}B_M]}{1 + \kappa d_M}$, where $d_M$ is the number of non-zero eigenvalues of $B'_M$.*

For a finite $\kappa$, these results suggest that collapses can still happen. For VICReg, $B = -A_0$, and the complete collapse can happen when $\kappa \ll \|A_0\|/c\|\Sigma\|$ – this explains the experimental observation of collapses for small values of $\kappa$ in VICReg loss (Bardes et al., 2021).

Lastly, to understand normalization, we are interested in the case of $\kappa \to \infty$. Combining Proposition 3 and 4, we have proved the following theorem, showing that the asymptotic solution converges to a form independent of $\kappa$.

**Theorem 2.** *Let $W_\kappa$ be a stationary point of Eq. (10) at fixed $\kappa$. Then,*

$$\lim_{\kappa \to \infty} W_\kappa^T W_\kappa = \frac{1}{2} \Sigma^{-1} \left[ B_M + \frac{2c - \text{Tr}[\Sigma^{-1} B_M]}{d_M} \Sigma_M \right] \Sigma^{-1}. \tag{11}$$

The correction term $\frac{2c - \text{Tr}[\Sigma B_M]}{d_0} \Sigma_M$ emerges as a result of applying normalization. The effect can be easier to understand if we write the solution as

$$W^T W = \frac{1}{2} \left[ \Sigma^{-1} B_M - \frac{\text{Tr}[\Sigma^{-1} B_M]}{d_M} M + \frac{2c}{d_M} \right] \Sigma^{-1}, \tag{12}$$

where we have used the relation $\Sigma_M \Sigma^{-1} = M$. Note the term in brackets: it subtracts the average eigenvalue of $\Sigma^{-1} B_M$ from $\Sigma^{-1} B_M$ and shifts the remaining eigenvalues positively by $2c/d_M$. Because the eigenvalues of $WW^T$ must be positive, the following condition must hold for all solutions:

$$\lambda_i + 2c/d_M > \bar{\lambda}, \tag{13}$$

where $\lambda_i$ are the eigenvalues of $\Sigma^{-1} B_M$ and $\bar{\lambda}$ is its average. Namely, for the $i$−th dimension not to collapse, it must be smaller than the average eigenvalues by at most $2c/d_M$. Any smaller eigenvalues must collapse. Compared to the case without normalization, normalization makes collapses dependent on the *relative* strength of each feature and augmentation. In the following discussion, we let $c = 1$ to simplify the discussion. We present a detailed analysis of this condition in Section D.1. One finds that the condition for collapse becomes heavily dependent on the data structure, and there are cases where collapses become harder, and there are cases where collapses become much easier. Importantly, it also becomes the case that a sufficiently strong augmentation can always cause a collapse in the corresponding subspace.

**Effect of Bias**. Lastly, we study the effect of explicitly having a bias term: $Wx \to Wx + b$. First of all, when there is no normalization, the bias term does not affect the solution because the loss landscape is invariant to a translation in the learned representation. However, this effect dramatically changes if we apply normalization at the same time. This is because normalization removes the translation symmetry of the effective loss, and the trivial solution $W = 0$, $b = 1$ becomes the simplest way to achieve the norm−1 constraint. Our result shows that the addition of bias dramatically affects the stationary points.

**Theorem 3.** *Let $f(x) = Wx + b$ and $\mathbb{E}[x] = 0$. Then, all stationary points $W$ satisfy Eq. (9), subject to the constraint that $\text{Tr}[W^T \Sigma W] \leq c$.*

Namely, the solution reverts to the case where there is no normalization at all, except that the norm of the solution can no longer be larger than $c$. This upper bound can make collapses much easier to happen. For example, if $c < (a_i - c_i)/(a_i + c_i)$ for all $i$, a complete collapse can happen despite normalization. When $c = 1$ and $c_i \ll a_i$, $\rho \approx d_M/2$ and the constraint indicates that $d_M \leq 2$: when the augmentation is very weak, there are at most 2 nontrivial subspaces. This is too restrictive for learning a meaningful representation, which helps us understand why dimensional collapse can harm learning in practice. The fact that simple normalization cannot prevent collapse has been noticed for a while for the simplest case of a cosine-similarity loss, and our result explains why previous works have tried to introduce asymmetry to cosine similarity to avoid collapses (Grill et al., 2020; Chen and He, 2021).

**Relevant Loss Functions**. Having developed a framework for understanding normalization, we show that other common loss functions in SSL can also be written in the form given in Eq. (3). The spectral contrastive loss (SCL) (HaoChen et al., 2021) reads

$$L_{SCL} = -2\mathbb{E}[f(x)^T f(x')] + \mathbb{E}[(f(x)^T f(\chi))^2] + const. \qquad \text{s.t. } \|f(x)\|^2 = 1. \tag{14}$$

Let $f(x) = Wx$ be linear, the distributions are zero-mean Gaussian, and ignore the normalization. This loss function becomes

$$L_{SCL} = -2\text{Tr}[WCW^T] + \text{Tr}[W\Sigma W^T W\Sigma W^T]. \tag{15}$$

When normalization exists, we can apply the result in Section 3.2. By our argument, there is no collapse in this loss function. The difference with InfoNCE loss is that the learned feature spreads along the directions of the augmentation $C$, not along the directions of the feature $A_0$.

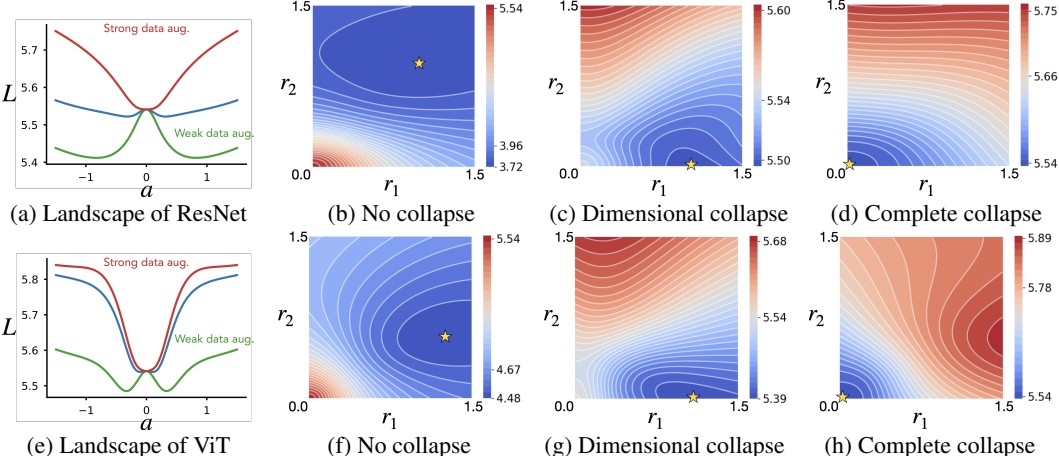

Figure 2: **Landscape of Resnet18 (upper) and vision transformers (lower) on CIFAR10 with SimCLR qualitatively agrees with our linear theory.** (a) Training objective $L$ as a function of a rescaling of the last layer $W \to aW$. (b-d) $L$ as a function of a $2d$ rescaling of the last layer where the data augmentation strength is (b) small, (c) intermediate, and (d) strong. Red indicates areas of high loss, blue indicates areas of low loss, and stars locate local minima. The use of data augmentation changes the stability of the origin, a qualitative change that leads to different types of collapses in qualitative agreement with our linear theory (cf. Figure 1). Additionally, we also notice the same qualitative changes of landscape in simpler nonlinear models (see Appendix A). (e-h) are the same setting but for ViT.

The case of Barlow Twin (BT) (Zbontar et al., 2021) is similar. While the fourth-order term of BT is much more complicated due to the imbalance created by the $\lambda$ term. The second-order term can be identified easily: $L_{BT} = -2\text{Tr}[W\Sigma W^T] + O(\|W\|^4)$. This also does not collapse. A difference between the SCL loss and InfoNCE is that the learned representation has a spread that aligns with the combination of the feature and the augmentation strength.

## 4    IMPLICATIONS

In this section, we explore some theoretical and practical implications of our results. In Appendix Section A, we also present numerical simulations that directly validate the predictions of the theory.

### 4.1    RELEVANCE TO NONLINEAR MODELS

An important question is how much of the analysis is relevant for deep nonlinear models in general. In fact, the loss landscape we have studied is quite close to the most general landscape one can have. Let $L(f(x))$ be a general SSL loss function for data point $x$. The quality of the learned representation should be independent of the population-level orientation of the representation. Therefore, the loss function should satisfy a rotational invariance. Namely, for any rotation matrix $R$, $L(x) = L(Rf(x))$; this rotational invariance implies that the loss should expand as $L(f(x)) = af(x)^T f(x) + b[f(x)^T f(x)]^2 + O(f(x)^6)$. Note that all the odd-order terms of $f(x)$ vanish due to the rotational symmetry. Substituting $f(x) = W\phi(x)$ in the loss function, we obtain a very general form of landscape that $W$ obeys:

$$L(W, \phi) = \text{Tr}[W^T W A] + \sum W_{im} W_{jm} W_{kn} W_{ln} Z_{ijki}, \tag{16}$$

where $A$ and $Z$ are dependent on $\phi$. Note how all the examples we have studied take this form. For $W$, its collapse entirely depends on the stability of the matrix $A$. Thus the study of the stability of the matrix $A$ becomes crucial for our understanding. To illustrate, we train a Resnet18 on CIFAR10 with the SimCLR loss with normalization and with weight decay strength $10^{-3}$ until convergence to obtain the converged weights $W^*$. The representation has a dimension 128. We rescale the weight matrix of the last layer $W^*_{\text{last}}$ by a factor $a$ and compute the loss as a function of $a$. See Figure 2-a. We then partition the singular values of $W^*_{\text{last}}$ into the larger half and the smaller half. We rescale the larger half by a factor $r_1$ and the smaller half by $r_2$. We plot the loss as a $2d$ function of $(r_1, r_2)$ in Figure 2. We also perform experiments for vision transformers (ViT) in the lower row (Dosovitskiy et al., 2020). In all cases, the landscape features qualitative changes comparable to those in Figure 1.

**A connection to Landau theory in physics.** Those familiar with statistical physics should note that the proposed theory is analogous to the Landau theory of second-order phase transitions. When treating the loss function as the free energy, the square root of the eigenvalues $\sqrt{\lambda}$ of $W^T W$ are the order parameters of the system, and the phase transitions happen when $\lambda$ turns from 0 to positive. These transitions (collapses) happen because of *symmetry breaking* (Landau and Lifshitz, 2013): the loss function (2) is symmetric in the sign of $W$. Yet, for any nontrivial learning, $W$ must be nonzero; thus, a symmetry breaking of the sign of $W$ needs to happen for learning. The recent work by Ziyin and Ueda (2022) suggested how symmetry breaking around the origin and Landau theory could explain various types of collapses in deep learning. Therefore, the dimensional collapse could be related to neural collapses in supervised learning (Papyan et al., 2020; Ziyin et al., 2022a) and posterior collapse in Bayesian deep learning (Wang and Ziyin, 2022). Because second-order phase transitions should come with the divergence of the correlation function, one might also wonder what is "divergent" in the SSL problem. Here, the learning time scale for the collapsing dimension is divergent at the critical point because the second-order term vanishes in this direction, and so the dynamics are effectively frozen along this direction.

## 4.2 Robustly inducing good collapses

Contrary to previous works, a recent work (Cosentino et al., 2022) has suggested that dimensional collapse can be beneficial and significantly improve the generalization performance of the model. This observation raises a question. How can dimensional collapse be beneficial and how can it be induced? In the following, we first introduce $\beta$-InfoNCE, which can adjust the degree of dimensional collapse, and analyze the collapse behavior to elucidate the mechanism of *task-alligned collapse*.

**Adjusting the degree of dimensional collapse with $\beta$-InfoNCE.** Despite the potential benefit, existing SSL loss functions cannot robustly induce dimensional collapse. InfoNCE is insufficient to induce a collapse, and the collapse induced by SimCLR depends on a vanishingly small parameter $1/N$. One thus wonders whether there is a loss function that allows us to induce collapsing behavior in a more predictable matter so that one might controllably extract some benefits from collapse. Our result suggests that one way to directly control collapses is through the strength of the competition for the model Hessian at the origin. For InfoNCE, one way to achieve this is to weigh the entropy term by a general factor $\beta$:

$$\mathbb{E}_x \left\{ \frac{1}{2} |f(x) - f(x')|^2 + \beta \log \mathbb{E}_\chi \left[ \exp \left( -\frac{1}{2} |f(x) - f(\chi)|^2 \right) \right] \right\}.$$

Due to its similarity with the $\beta$-VAE in Bayesian learning, we call it the $\beta$-InfoNCE. The leading term in the loss function becomes

$$-\mathrm{Tr}[W(A_0 - (1 - \beta)C)W^T].$$

When $1 - \beta > 0$, the augmentations $C$ pull the representation towards zero. When the augmentation is as strong as the feature variations, a collapse happens. One can thus introduce collapse by setting $\beta$ to be sufficiently small. When $1 - \beta < 0$, the augmentations push the weights away from the origin along its direction, resulting in no collapse at all: When one really wants to avoid collapse, one can use a rather large $\beta$; $\beta = 1$ is thus at the boundary of this bifurcating behavior. We note that existing loss functions often do not have a parameter that is directly controlling the collapse behavior (see Table 1). The $\beta$ parameter here directly controls the level of difficulty of collapse.

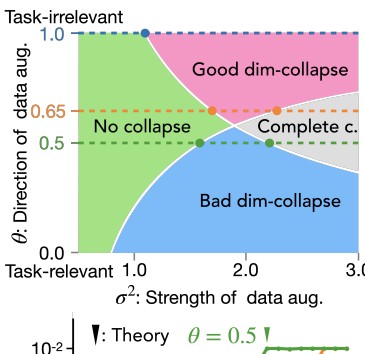

Figure 3: **Top**: Phase diagram of representational collapses. **Bottom**: $\beta$–InfoNCE with $\beta = 0.5$. The generalization error of a downstream regression task where the data augmentation (1) is isotropic and noninformative or (2) aligns with the style. We see that the performance worsens as collapses happen for the noninformative augmentation and improves as the collapse happens for the style-targeting augmentation.

**Achieving invariance with dimensional collapse.** Here, we closely study an illustrative minimal example to demonstrate how collapses can be beneficial. Consider the following structured data generating process where the input features can be separated into two sets: (1) a task-relevant set with dimension $d_c < d_0$ and (2) a task-irrelevant set: $x = (x_1, ..., x_{d_c}, ..., x_{d_0})$. Our result suggests

a precise way to remove the irrelevant features from the learned representation. For the purpose of causing a robust collapse, we use the $\beta$-InfoNCE with $\beta = 1/2$. For illustration, we consider the simple case $d_c = 1$ and $d_0 = 2$. For any input $x = (x_1, x_2)$, the label is generated as a linear function of $x_1$: $y = cx_1$.

Correspondingly, we consider a structured data augmentation $x = \hat{x} + \sigma R\xi$, where $R \in \mathbb{R}^{d_0 \times d_0}$ is $R = diag(\sqrt{1 - \theta}, \sqrt{\theta})$, where $\theta \in [0, 1]$. The parameter $\sigma$ controls the overall strength of the augmentation, and $\theta$ controls the orientation of the strength. When $\theta = 0.5$, we have an uninformative isotropic noise that has often been used in practice. When $\theta = 1$, the augmentation is only on the task-irrelevant feature, and when $\theta = 0$, the augmentation is only on the content. Since the prediction target only depends on the content, we want to learn a representation invariant to the style. For the downstream regression task, we use the learned representations $z := f(\hat{x})$ to train a ridge linear regressor that minimizes $\min_G \mathbb{E}_{\hat{x}}[\|Gz - y(\hat{x})\|^2] + 0.001\|G\|^2$. See Figure 3. The top panel shows the phase diagram of this problem with different combinations of the augmentation strengths and orientations. The bottom panel shows that collapses introduce phase-transition-like behaviors in the generalization performance and that a data augmentation aligning with the task-irrelevant dimension improves performance.

## 4.3 ROBUSTNESS TO DATA IMBALANCE

Our theory is not only relevant for understanding collapses but can also be used to understand how an SSL model encodes the feature. Liu et al. (2021) recently showed that compared with supervised learning, SSL techniques are relatively more robust to imbalanced datasets that have disproportionately represented minority subgroups. As another application of our analysis, we illustrate the robustness of different techniques is not equal. As we have seen, the learned model $W^T W$ has eigenvalues that, to the leading order, are proportional to the Hessian $B$, which is different for each loss function. As previously summarized in Table 1, for InfoNCE and SimCLR, the learned model aligns

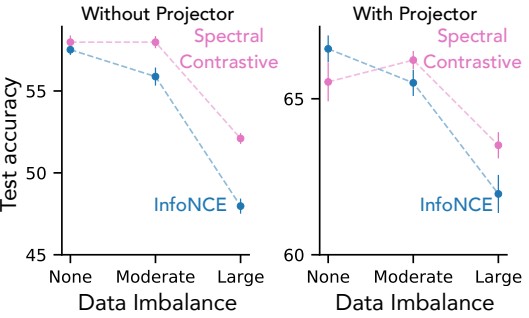

Figure 4: **Spectral Contrastive loss (SCL) is more robust against data imbalance than InfoNCE.** We train SimCLR and SCL ResNet-12 models on imbalanced versions of CIFAR-10. We see that SCL is more robust than SimCLR, as suggested by our theory. These results are especially pronounced when there is no projector head.

with the eigenvalues of the data covariance $A_0$, which varies hugely as different classes of a dataset become more and more imbalanced. In comparison, the model trained with SCL aligns purely with the augmentation covariance $C$, which is independent of the data imbalance. This suggests that the SCL landscape can be less dependent on data and thus more robust against data imbalance. See Figure 4. More experimental details are given in Appendix C.

## 5 CONCLUSION

In this work, we approached the problem of collapses in SSL from a loss landscape perspective. We analytically solved an effective landscape that can be extended to understand the effect of normalization. Our result suggests that dimensional collapse can be well understood in the minimal setting and is something neutral to learning on its own. With the help from the theory, we also showed that when task-irrelevant dimensions are targeted, dimensional collapse can result in improved performance, whereas an uninformative noise will (without good luck) leads to collapses in the dimensions that are relevant to the task. It is thus important for practitioners to devise targeted data augmentation mechanisms that incorporate the correct domain knowledge. Also, we advocated the thesis that the local geometry of the loss landscape around the origin is an essential component for understanding collapses, and this should invite more future work to understand the landscape around the origin.

The limitation of our work is clear; our result only identifies the causes of the collapse that can be directly attributed to the low-rank structure of the local minima of the landscape. One possible alternative cause of the collapse is dynamics. For example, having a large learning rate and small batch can sometimes cause a convergence towards the saddle points in the landscape (Ziyin et al., 2022b), which, as we have shown, are the collapsed solutions. Investigating the role of dynamics in the collapse is thus a crucial future problem.

ACKNOWLEDGEMENTS

This work was supported by a KAKENHI Grant No. JP18H01145 from the Japan Society for the Promotion of Science. Ziyin has been financially supported by the JSPS fellowship and thanks Zihan for the generous help during the writing of this paper. ESL was partially supported via NSF under the award CNS-2008151.

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

## A    ADDITIONAL NUMERICAL RESULTS

In this section, we validate our theory with numerical results. Unless specified otherwise, the dimension of the learned representation is set to be equal to the input dimension: $d_0 = d_1$.

**No Collapse for InfoNCE**. We showed that there is no collapse at all for the vanilla InfoNCE, no matter how strong the augmentation is. Our result implies that the smallest singular of the model $W$ scales as $\sigma^4$ where $\sigma^2$ is the strength (namely, the variance) of the augmentation. See the left panel of Fig. 5. We use the vanilla InfoNCE loss defined in (1) with a linear model. The training set is sampled from $\mathcal{N}(0, I_{32})$. The training proceeds with Adam with a learning rate of $6e-4$ with full batch training for 5000 iterations. We use a simple diagonal Gaussian noise with variance $\sigma^2$ for data augmentation. We see that the singular values scale as $\sigma^4$ and never vanishes, as the theory predicts.

**Nonrobust Collapses of Weighted InfoNCE**. We now demonstrate that, as the theory predicts, collapses of weighted InfoNCE depend strongly on the dataset size. We use the same dataset and training procedure as the previous experiment. We set $\alpha = 0.1$ and change the size of the training set. Theory suggests that for a collapse in the $i$−th subspace to happen, the size of the dataset needs to obey

$$N > \frac{a_i}{c_i(1-\alpha)} := N_{crit}. \tag{17}$$

See the middle panel of Figure 5. We show the smallest three eigenvalues of $W^T W$ (roughly having similar magnitudes), and the critical dataset size for the smallest eigenvalue. We see that the theoretical threshold of collapse agrees well with where the collapse actually happens.

**Collapses in $\beta$-InfoNCE**. With $\beta < 1$, one can cause collapses in a predictable and controllable way. In this experiment, we let $d_0 = 5$ and we plot all five eigenvalues of $W^T W$ as we increase the strength of an isotropic augmentation. As the numerical results show, collapses happen at the points predicted by the theory.

**Normalization Causes Dimensional Collapse**. We also plot the three smallest eigenvalues of $W^T W$ when we apply the standard representation normalization in practice: $f(x) \rightarrow f(x)/\|f(x)\|$. To facilitate comparison, we also use the same dataset and training procedure as before. See Figure 6. We see that normalization does cause a collapse in the smallest eigenvalues at an augmentation strength much smaller than the feature variation.

## B    LANDSCAPE OF A NONLINEAR MODEL

In this section, we plot the landscape of the layer of nonlinear models on the same synthetic dataset we outlined in the previous section. We train a three-layer nonlinear network with output dimension 2 with SGD until convergence. We then rescale the optimized weight of the last by a factor $a$: $W_{last} \rightarrow aW_{last}$ and plot the loss function along this direction. See the top panel of Figure 7 for

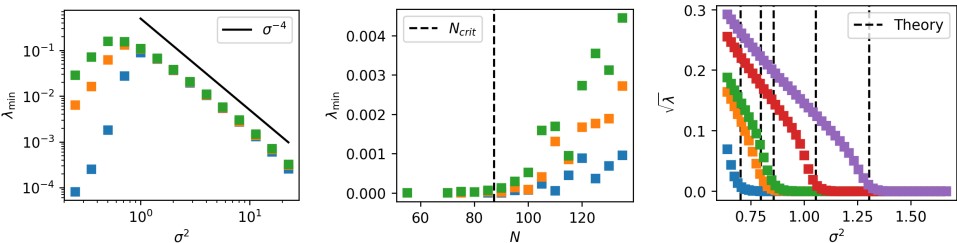

Figure 5: The three smallest singular values of $W^T W$ as a function of the augmentation strength. We see that our effective landscape theory around the origin accurately captures collapses in learning. **Left**: Vanilla InfoNCE . As the theory suggests, the singular values scale as $\sigma^4$ and do not vanish for any finite value of $\sigma$. **Mid**: Weight InfoNCE. $\alpha = 0.1$, $\sigma = 5$. Collapse happens at the critical dataset size predicted by the theory. **Right**: (Sqrt) Eigenvalues of $WW^T$ in $\beta$-InfoNCE. The collapses can be well controlled.

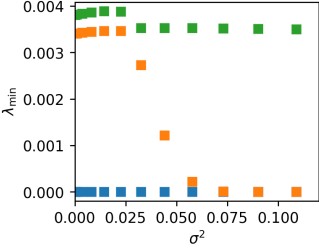

Figure 6: A collapse happens easily when the learned representation is normalized. The smallest eigenvalues of $A_0$ are roughly 0.2, and the collapse happens much before the noise reaches this strength.

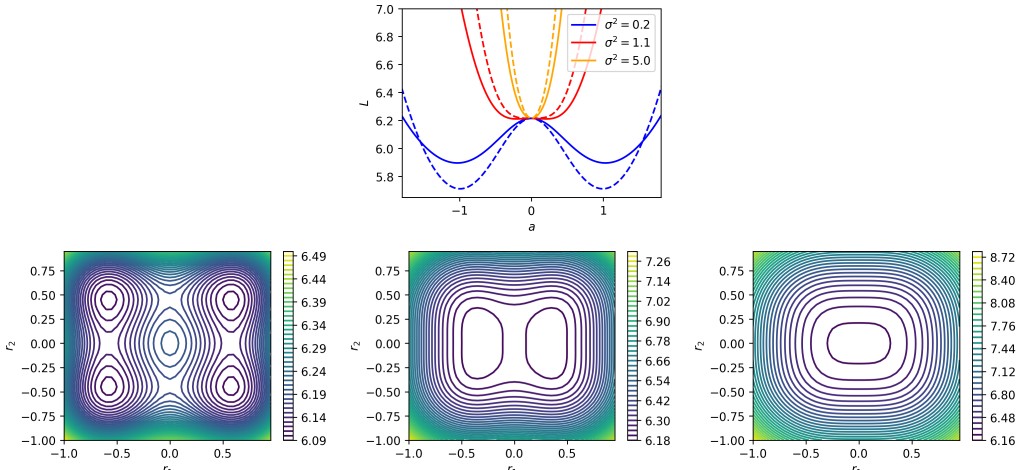

Figure 7: The Landscape of nonlinear models is very similar to the landscape of linear models (cf. Figure 1). **Top**: $1d$ projection of the landscape of a two-layer tanh and ReLU network. **Bottom Left**: the landscape of a 2D projection of the last layer of a nonlinear model with a weak augmentation. **Middle**: with intermediate augmentation. **Right**: with strong augmentation.

both the tanh and the ReLU nonlinearity. We then rescale the two rows of the weight matrix of the model by $r_1$ and $r_2$ respectively: $W = (w_1, W_2)^T \rightarrow (r_1 w_1, r_2 w_2)$. We see that the landscape of the model is qualitatively the same as that of the linear models, shown in Figure 1.

## C   SETUP FOR IMBALANCED DATA EXPERIMENTS

*Creating an Imbalanced Dataset:* For our experiments measuring the influence on imbalanced datasets on SSL training, we use CIFAR-10 by sampling 20000 samples out of the 50000 training samples. The sampling process is described by a Dirichlet distribution and is often used to analyze effects of heterogeneity and data imbalance in Federated Learning problems (Hsu et al., 2019). Specifically, a small value of the distribution parameter yields a highly imbalanced dataset, while a large value yields a perfectly balanced dataset. We evaluate our models in three scenarios, for which we report below the number of samples per class:

- High imbalance: [4890, 87, 5000, 0, 74, 0, 0, 212, 4788, 4947]
- Medium imbalance: [4268, 4296, 1741, 420, 945, 161, 4633, 1015, 131, 2386]
- No imbalance: [2000, 2000, 2000, 2000, 2000, 2000, 2000, 2000, 2000, 2000]

*Training Setup:* We use ResNet-12 models as the backbone for all experiments due to computational constraints. SimCLR augmentations (Chen et al., 2020a) are followed, except for a reduced strength of resized cropping from 0.2 to 0.5. All training involves a standardly used cosine decay learning

rate schedule, starting at 0.03 and decaying to 0.001. When a projector module is used, it involves a two-layer MLP with hidden dimension of 512 and BatchNorm layer in between. We use SGD for optimization and perform the standardly used linear evaluation protocol for measuring the quality of the final representation. For training the linear layer, we use an initial learning rate of 10 and decay it to 0.01 with a cosine schedule. We note linear evaluation is used for supervised models as well, following the practice advocated by Liu et al. (2021).

## D ADDITIONAL THEORETICAL CONCERNS

### D.1 COLLAPSE CONDITION FOR NORMALIZATION

The important condition for collapse in Eq. (13) can be better understood by considering the extreme cases. First of all, note that the eigenvalues of $\Sigma B_M$ are bounded between $-1$ and $1$

$$-1 \leq \frac{a_i - c_i}{a_i + c_i} \leq 1, \tag{18}$$

and $-1$ is achieved when $c_i \gg a_i$, and $1$ is achieved when $a_i \gg c_i$.

When the augmentation is negligibly small, $\Sigma^{-1} B_M \approx M$, and $\lambda_i \approx \bar{\lambda} = 1$, the condition thus becomes

$$\frac{2}{d_M} > 0, \tag{19}$$

which always holds. Thus, a sufficiently small augmentation will never cause collapse. Next, when we apply very strong augmentation to the $j$-th subspace and zero augmentation to the others, the condition for the non-augmented spaces becomes

$$1 + \frac{2}{d_M} > \frac{d_M - 2}{d_M}, \tag{20}$$

meaning that the collapse will not happen. For the $j$-th space, the condition is

$$-1 + \frac{2}{d_M} > \frac{d_M - 2}{d_M} (\Longleftrightarrow) \frac{4}{d_M} > 2, \tag{21}$$

which is only possible when $d_M = 1$, namely, the strongly augmented space is the only space that does not collapse. This is reasonable when the original data is rank-1 because the normalization will ensure that this space does not collapse, but when the original data is not rank-1, this stationary point will be a saddle and will not be preferred by gradient descent. In different word, a strong enough augmentation will cause a collapse in the corresponding subspace, as is the case without normalization.

It is also interesting to note that having $c_i \geq a_i$ is no longer sufficient to cause a collapse. For example, let $c_1 = 0$ and $c_j = a_j$ for $j \neq 1$. The condition for $j \neq 1$ becomes

$$\frac{2}{d_M} > \frac{1}{d_M}, \tag{22}$$

which always holds. At the same time, it does not mean that collapsing has become harder in general. For example, it is also possible for $c_i < a_i$ to cause a collapse. Suppose we add a weak augmentation only to the first subspace such that $a_i - c_i = \epsilon > 0$, the condition for this dimension to not to collapse is

$$\frac{\epsilon}{a_i + c_i} + \frac{2}{d_M} > \frac{d_M - 1 + \epsilon}{d_M}, \tag{23}$$

which can be violated whenever $\epsilon < \frac{(a_i + c_i)(d_M - 3)}{a_i + c_i + d_m}$. Namely, in some cases, normalization can in fact facilitate collapse.

# E   PROOFS

## E.1   PROOF OF PROPOSITION 1

*Proof.* The second term in Eq. (3) can be written as

$$\mathrm{Var}[|W(x-\chi)|^2] = \mathbb{E}\left[(\mathrm{Tr}[W(x-\chi)(x-\chi)^TW^T])^2\right] - \mathbb{E}\left[\mathrm{Tr}[W(x-\chi)(x-\chi)^TW^T]\right]^2 \tag{24}$$

$$= [first\ term] - 4\mathrm{Tr}[W(A_0+C)W^T]^2 \tag{25}$$

$$= [first\ term] - 4\mathrm{Tr}[W\Sigma W^T]^2, \tag{26}$$

where we have used the definition $\Sigma = A_0 + C$. The first term is

$$[first\ term] = \mathbb{E}\left[(\mathrm{Tr}[W(x-\chi)(x-\chi)^TW^T])^2\right] = 4\mathrm{Tr}[W\Sigma W^T]^2 + 8\mathrm{Tr}[W\Sigma W^TW\Sigma W^T]. \tag{27}$$

Combining the above expressions, we see that Eq. (3) can be written as

$$L = -\mathrm{Tr}[WBW^T] + \frac{1}{8}\mathrm{Var}[|W(x-\chi)|^2] \tag{28}$$

$$= -\mathrm{Tr}[WBW^T] + \mathrm{Tr}[W\Sigma W^TW\Sigma W^T]. \tag{29}$$

This finishes the proof. □

## E.2   PROOF OF THEOREM 1

*Proof.* All stationary points have a zero gradient:

$$-2WB + 4W\Sigma W^TW\Sigma = 0. \tag{30}$$

Multiplying by $W^T$ on the left and $B^{-1}$ on the right,

$$W^TW = 2W^TW\Sigma W^TW\Sigma B^{-1} \tag{31}$$

$$(\Longleftrightarrow) \quad \Sigma^{1/2}W^TW\Sigma^{1/2} = 2\Sigma^{1/2}W^TW\Sigma W^TW\Sigma B^{-1}\Sigma^{1/2} \tag{32}$$

Defining $H \coloneqq \Sigma^{1/2}W^TW\Sigma^{1/2}$, we obtain

$$H = 2H^2\Sigma^{1/2}\Sigma B^{-1}\Sigma^{1/2}, \tag{33}$$

$$(\Longleftrightarrow) \quad H(I - 2H\Sigma^{1/2}B^{-1}\Sigma^{1/2}) = 0. \tag{34}$$

Because both $H$ and $\Sigma^{1/2}\Sigma B^{-1}\Sigma^{1/2}$ are symmetric, one can take the transpose of Eq. (33) to find that $H$ and $\Sigma^{1/2}B^{-1}\Sigma^{1/2}$ commute with each, which implies that $H$ has the same eigenvectors as $\Sigma^{1/2}B^{-1}\Sigma^{1/2}/2$.

Eq. (34) then implies that the eigenvalues of $H$ is either the inverse of that of $\Sigma^{1/2}B^{-1}\Sigma^{1/2}$ or zero. This implies that any stationary point of $H$ can be written in the form

$$H = \frac{1}{2}UM\Lambda U^T, \tag{35}$$

where $U$ is a unitary matrix, $\Lambda$ is diagonal matrix containing the eigenvalues of $\Sigma^{1/2}B^{-1}\Sigma^{1/2}$, and $M$ is an arbitrary (masking) diagonal matrix containing only zero or one such that (1) $M_{ii} = 0$ if $\Lambda_{ii} < 0$ and (2) contain at most $d^*$ nonzero terms. This then implies that the weight matrix $W$ satisfies

$$W^TW = \frac{1}{2}\Sigma^{-1/2}UM\Lambda U^T\Sigma^{-1/2}. \tag{36}$$

Lastly, when $\Sigma$ and $B$ commute, we can compactly write the result as

$$W^TW = \frac{1}{2}\Sigma^{-1}B_M\Sigma^{-1}, \tag{37}$$

where $B_M$ denotes the matrix obtained by masking the eigenvalues of $B$ with $M$. This finishes the proof. □

### E.3 PROOF OF PROPOSITION 2

*Proof.* For all stationary points, $W^T W$ commutes with $B$ and $\Sigma$, which means that at these stationary points, one can simultaneously diagonalize all the matrices and the loss function (3) can be written as

$$L = -\sum_{i=1}^{d^*} \lambda_i b_i + \lambda_i^2 s_i^2 \tag{38}$$

where $\lambda_i$, $b_i$, $s_i$ are the eigenvalues of $W^T W$, $B$, and $\Sigma$ respectively.

We can thus consider each $i$ separately. When $b_i > 0$, $\lambda_i = 0$ cannot be a local minimum because the local Hessian is $-b_i < 0$. When $b_i \leq 0$, the only stationary point is $\lambda_i = 0$. This sum covers at most $d^*$ summands, and so, at the local minima, $\lambda_i \neq$ if and only if $b_i > 0$, and so the number of non-zero eigenvalues of $W^T W$ is $\min(m, d^*)$. $\square$

### E.4 PROOF OF PROPOSITION 3

*Proof.* The regularization can be written as

$$R = [(\mathbb{E}_x \|Wx\|^2 - c)^2] \tag{39}$$
$$= \text{Tr}[W\Sigma W^T]^2 - 2c\text{Tr}[W\Sigma W^T] + c^2. \tag{40}$$

By Proposition 1, Eq. (10) reads

$$L = -\text{Tr}[WBW^T] + \text{Tr}[W\Sigma W^T W\Sigma W^T] + \kappa(\text{Tr}[W\Sigma W^T]^2 - 2\text{Tr}[W\Sigma W^T] + 1) \tag{41}$$
$$= -\text{Tr}[W(B + 2\kappa c\Sigma)W^T] + \text{Tr}[W\Sigma W^T W\Sigma W^T] + \kappa\rho^2. \tag{42}$$

The derivative of $\rho$ is

$$\frac{d}{dW}\rho = 4\rho W\Sigma. \tag{43}$$

The zero-gradient gradient is thus

$$-2W(B + 2\kappa c\Sigma - 2\kappa\rho\Sigma) + 4W\Sigma W^T W\Sigma = 0. \tag{44}$$

We can define $B' := B + 2\kappa c\Sigma - 2\kappa\rho\Sigma$ to see that this condition is the same as Eq. (30) in the proof of Theorem 1. The rest of the proof thus follows from the arguments. We thus arrive at the theorem statement:

$$W^T W = \frac{1}{2}\Sigma^{-1} B'_M \Sigma^{-1}. \tag{45}$$

We are done. $\square$

### E.5 PROOF OF PROPOSITION 4

*Proof.* Recalling that $\rho = \text{Tr}[W\Sigma W^T]$, we multiply $\Sigma$ from the right to both sides of the solution in Proposition 3 and take trace:

$$\frac{1}{2}\text{Tr}[\Sigma^{-1} B'_M] = \frac{1}{2}\text{Tr}[\Sigma^{-1}(B_M + 2\kappa(c - \rho)\Sigma_M)] \tag{46}$$
$$= \text{Tr}[W^T W\Sigma] \tag{47}$$
$$= \text{Tr}[W\Sigma W^T] = \rho. \tag{48}$$

The first line further simplifies to

$$\frac{1}{2}\text{Tr}[\Sigma^{-1} B_M] + \kappa(c - \rho)\text{Tr}[\Sigma^{-1}\Sigma_M] = \frac{1}{2}\text{Tr}[\Sigma^{-1} B_M] + \kappa(c - \rho)d_M, \tag{49}$$

where $d_M := \text{Tr}[M]$ is the number of nonzero eigenvalues of $B'_M$.

This gives an equation of $\rho$ that solves to

$$c - \rho = \frac{c - \frac{1}{2}\text{Tr}[\Sigma^{-1} B_M]}{1 + \kappa d_M}. \tag{50}$$

This proves the proposition. $\square$

# F  ADDITIONAL THEORETICAL CONCERNS

## F.1  CASE OF DATA-INDEPENDENT NON-GAUSSIAN AUGMENTATION

In the main text, we mainly considered the case when the noise is Gaussian. In this section, we consider a case where the noise is data-dependent and non-Gaussian. We show that the results we discussed in the main text still hold qualitatively. The general form of the loss function in Eq. (3) still applies:

$$L = -\text{Tr}[WBW^T] + \frac{1}{8}\text{Var}[|W(x-\chi)|^2].  \tag{51}$$

We consider a global rescaling augmentation for each datum $x$:

$$x = s\hat{x},  \tag{52}$$

where $s \sim exp(b)$ obeys an exponential distribution with mean $b$ and variance $b^2$. Note that even if $\hat{x}$ is Gaussian, the augmented data is no longer Gaussian. In particular, the augmentation now becomes data-dependent. This augmentation can also be seen as a structured, biologically plausible data augmentation that encourages the model to be scale-invariant, which is what Wien's law for biological perception demands (Dayan and Abbott, 2005): no matter whether an image is dark or bright, the content of the image is the same.

Under this augmentation, the noise covariance is dependent on $x$ and no longer Gaussian:

$$\mathbb{E}[xx^T] = 2b^2 A_0.  \tag{53}$$

We also obtain that

$$C = \mathbb{E}[(b-s)^2 xx^T] = b^2 A_0.  \tag{54}$$

The second term in Eq. (3) can be written as

$$\text{Var}[|W(x-\chi)|^2] = \mathbb{E}\left[(\text{Tr}[W(x-\chi)(x-\chi)^T W^T])^2\right] - \mathbb{E}\left[\text{Tr}[W(x-\chi)(x-\chi)^T W^T]\right]^2  \tag{55}$$

$$= [first\ term] - 4\text{Tr}[W(A_0+C)W^T]^2  \tag{56}$$

$$= [first\ term] - 4\text{Tr}[W\Sigma W^T]^2,  \tag{57}$$

where we have used the definition $\Sigma = A_0 + C$. The first term is

$$[first\ term] = \mathbb{E}\left[(\text{Tr}[W(x-\chi)(x-\chi)^T W^T])^2\right].  \tag{58}$$

However, for fixed rescaling factor $s_x$ and $s_\chi$, each $W(x-\chi)$ obeys a multivariate Gaussian distribution with variance $2(s_x^2 + s_\chi^2)WA_0$, and so we have

$$[first\ term] = \mathbb{E}_{s_x,s_\chi}[(s_x^2 + s_\chi^2)^2](4\text{Tr}[WA_0W^T]^2 + 8\text{Tr}[WA_0W^TWA_0W^T]),  \tag{59}$$

where $\mathbb{E}_{s_x,s_\chi}[(s_x^2 + s_\chi^2)^2] = 56b^4$. Combining terms, we obtain that

$$\text{Var}[|W(x-\chi)|^2] = 48b^2 \times 4\text{Tr}[WA_0W^T]^2 + 56b^4 \times 8\text{Tr}[WA_0W^TWA_0W^T].  \tag{60}$$

The loss function is thus:

$$L = -\text{Tr}[WBW^T] + 24b^2\text{Tr}[WA_0W^T]^2 + 56b^4\text{Tr}[WA_0W^TWA_0W^T].  \tag{61}$$

Note that this loss function is a special case of the loss function in Eq. (10) where $c = 0$ and $\kappa = 24b^2$ (and with a rescaled fourth-order term). As in the main text, $B$ is different according to different choices of loss functions. Because $B$ commute with $A_0$ by construction, one expects collapses to happen at locations predicted by Proposition 3 and 4 under suitable choices of parameters. Also note that the odd terms vanish as discussed, and so the local stability of the origin should decide the collapsing behavior of this situation.

This shows that collapse can also happen when the data augmentation is structured. We comment that the analysis in this section is minimal, and one important future direction is to provide more precise and insightful conditions of collapse under structured data augmentation.

