# OpenReview forum: "What shapes the loss landscape of self supervised learning?"
_ICLR.cc/2023/Conference — ICLR 2023 poster_

### Official Review · Reviewer_Q5CS · 2022-10-22

**Confidence:** 4
**Correctness:** 3
**Technical Novelty And Significance:** 3
**Empirical Novelty And Significance:** 3
**Recommendation:** 6

**Clarity, Quality, Novelty And Reproducibility:**

Clarity: the paper is clear with strong claim about the collapse in SSL.

Quality: The quanlity of this paper is good.

Novelty: neural collapse is not new, but this paper tries to fill the gap about conflict claim about collapse.

Reproducibility: It not clear since the code is not available.

**Strength And Weaknesses:**

Strength

1. Existng work have conflict opinion about the collapse in SSL. This work try to fill this gap by studying the geometry of the SSL.

2. This work shows that the interplay between data variation and data augmentation determines the geometry of the loss.

3. They found the geometry of the loss explains when dimensional collapse can be helpful and why certain SSL
losses are robust against data imbalance, but not the other.

Such findings may help design a better loss in SSL.


Weakness:
1. The work may rely on the assumption that the augementation is x = x +e, and all the data and noise must be Gaussian also, the nerual is linear. This assumption may be strong, since they are many non-noise based augementation. It is not clear if other data augumentation still has similar results as in this work.

2. If the data is imbalance as shown in 4.3 How the entire analysis appiled to this case, like if data is not Gaussian?

3. The code is not available, the reproducibility is unclear.

**Summary Of The Paper:**

This paper investigates the landscape causes of collapse in self-supervised learning.


**Summary Of The Review:**

In general, Although the paper has some strong assumption, it still gives some insight about how the collapse happens in SSL.

---

> ### Author Response · Authors · 2022-11-15
> **author reply**
>
> Thank you for your constructive feedback. We are pleased to hear that you found our submission clear, of good quality, and fills a gap in our understanding of representational collapse in SSL. Below, we fully address your concerns by newly adding analysis for non-gaussian data augmentation and providing a code for validating the main theoretical result.
>
> **"The work may rely on the assumption that the augementation is x = x +e, and all the data and noise must be Gaussian also, the nerual is linear. This assumption may be strong, since they are many non-noise based augementation. It is not clear if other data augumentation still has similar results as in this work."**
>
> Thanks for this constructive question. To reflect your valuable feedback, we now include a new section (Section F) to study an example with a structured noise that encourages a scale-invariance in the learned representation. The result shows that the main results are still qualitatively the same here, robustifying our results. Furthermore, empirical confirmation of our theory’s qualitative prediction in the case of data imbalance strengthens the robustness of our analysis in practical settings with non-gaussian data augmentation.
>
> **"If the data is imbalance as shown in 4.3 How the entire analysis applied to this case, like if data is not Gaussian?"**
>
> As discussed in the paper (section 4.1), the result can be qualitatively applied to a generic non-Gaussian case because the collapse phenomenon is determined by the second-order term in the loss function, whereas the non-Gaussianity only affects the fourth-order term and above. Thus, when the data is imbalanced, its data covariance will also be imbalanced. Specifically, the eigenvalues corresponding to the class with fewer data will be smaller. This directly influences the stability and geometry of the loss function around the origin, which, in turn, determines the collapse phenomenon, as the proposed theory implies.
>
> **"The code is not available, the reproducibility is unclear."**
>
> We now provide a preliminary code to validate the main theoretical results. We will provide more extensive polished code and add annotations after publication.
>
> **Summary**: Thank you again for the constructive feedback that led us to develop a new section on the analysis of non-Gaussian data augmentation. Together with the numerical analysis with non-Gaussian data augmentation in a practical setting with data imbalance and the extensive code base we now provide for reproducibility, we believe we have now fully addressed your concerns. In light of those updates, we would appreciate it if you could consider increasing the rating score to support the acceptance.

---

### Official Review · Reviewer_m5m1 · 2022-10-25

**Confidence:** 3
**Correctness:** 4
**Technical Novelty And Significance:** 3
**Empirical Novelty And Significance:** 2
**Recommendation:** 6

**Clarity, Quality, Novelty And Reproducibility:**

The paper has moderate clarity, quality, novelty, and poor reproducibility. There is no source code provided or detailed experimental settings available to reproduce the complex analysis in the paper.

**Strength And Weaknesses:**

Pros: The authors conduct a solid and comprehensive theoretical analysis of the SSL landscape. For example, in Section 3.1, the authors show that the landscape for a class of situations in self-supervised contrastive learning can be reduced to an effective form in Eq. (3). In Proposition 1, the authors show that the variance term of the loss takes a specific form when the data is Gaussian. Moreover, in Section 4 and Appendix A, the authors illustrate some theoretical and practical implications of their analytic results.

Cons: Although focused on analyzing the reasons for the collapse in SSL, this work cannot account for all the SSL collapses but only identifies the results that can be directly attributed to the low-rank structure of the local minima of the landscape. The paper's numeric analysis is limited. The language and organization of the draft have space to be improved. The paper is not easy to understand due to the lack of intuitive explanation.

The authors are suggested to improve the draft by improving the points mentioned in the Cons to improve the paper.

**Summary Of The Paper:**

In this work, the authors analyze the problem of collapses in SSL from a landscape perspective. They solved a landscape that can be extended to understand the effect of normalization. Their result suggests that dimensional collapse can be explained in the minimal setting and is something neutral to learning on its own. They showed that when task-irrelevant dimensions are targeted, dimensional collapse can result in dramatically improved performance, whereas an uninformative noise will lead to collapses in the dimensions that are relevant to the task. The authors believe that It is thus important for practitioners to devise targeted data augmentation mechanisms that incorporate the correct domain knowledge. The proposed theory can serve as a theoretical foundation and baseline of any advanced theory of collapses because a correct theory should agree with our results when restricting to the case of a linear model. The authors advocated the thesis that the local geometry of the loss landscape around the origin is an essential component for understanding collapses, and this should invite more future work to understand the landscape around the origin.



**Summary Of The Review:**

Please refer to the above sections to find the points that can be revised to improve the paper.

---

> ### Author Response · Authors · 2022-11-15
> **author reply**
>
> Thank you so much for taking the time to carefully read through our paper, and we are pleased that you found our submission to be “a solid and comprehensive theoretical analysis.” We now fully address your valuable feedback by providing more intuitive explanations and extensive numerical analysis.
>
>
> **"Although focused on analyzing the reasons for the collapse in SSL, this work cannot account for all the SSL collapses but only identifies the results that can be directly attributed to the low-rank structure of the local minima of the landscape. "**
>
> Thank you for this comment, but could you clarify your statements so that we can precisely understand and address your concern? Specifically: when you say, “this work cannot account for all the SSL collapses,” this statement is without a reference. Could you provide a reference to the SSL collapses that our theory cannot account for and that you believe to be important? In our understanding, the only type of SSL collapse is the type that comes with a low-rank structure. Can you point out any other type of SSL collapse that has been studied?
>
> That being said, the only limitation here is that we do not consider the low-rank structure that is attributable to the implicit bias of SGD, but this type of bias appears broadly in problems that are not SSL-relevant. In this regard, our paper has tackled all the known types of collapses that are specific to SSL.
>
>
>
> **"The paper's numeric analysis is limited. "**
>
> Thank you for this feedback. To address your concern, we now included extensive numerical experiments to thoroughly test the validity of our theory, including
> - Landscapes of ResNet18 trained with varying strengths of data augmentation. (Fig. 2, top)
> - Landscapes of Vision Transformer trained with varying strengths of data augmentation. (Fig. 2, bottom)
> - ResNet models trained on datasets with varying degrees of imbalance, confirming our theoretical predictions on the Spectral Contrastive Loss over InfoNCE. (Fig. 4)
> - Validating “quantitative” predictions of our theory in a three-layer “nonlinear” network (App. Fig. 5)
> - Validating predictions of our theory in a three-layer nonlinear network “with normalization” (App. Fig. 6)
> - Thorough numerical comparison of landscapes between linear and nonlinear models, validating our results. (App. Fig. 7)
>
>
>
> **"The language and organization of the draft have space to be improved. The paper is not easy to understand due to the lack of intuitive explanation."**
>
> Thank you for this feedback. We have now added new experiments, including one with Vision Transformer, through which we effectively translate takeaways of our theoretical analysis into practically relevant settings. While our paper is theoretical in nature, we highlight that we made efforts to communicate intuitions and takeaways effectively:
> - Intuitive schematic illustration of how the strength of augmentation and data variation controls the landscape. (Fig. 1)
> - Table summarizing the results of our analysis for 7 distinct SSL loss functions widely used in practice. (Table 1)
> - Visualization of loss landscapes for ResNet18 and CIFAR10 (Fig. 2)
> - Application of our theory to a simple and interpretable setting (Sec. 4.2 and Fig. 3)
>
> Please let us know if there is any part of our paper that you think would benefit from further intuitive explanations!
>
>
> **Summary**: Thank you for providing feedback that has greatly improved our manuscript. As we discussed above, our updated draft now includes (i) thorough empirical validations of our theoretical analysis in over six settings, (ii) an extensive code base so that all the results are fully reproducible, (iii) intuitive explanations through extensive visualizations and careful analysis of interpretable models. Please let us know if any further clarification would help, we are happy to implement them! We have hopefully addressed all of your concerns and would appreciate it if you could increase the rating and confidence score based on the clarifications

---

### Official Review · Reviewer_ffca · 2022-10-29

**Confidence:** 4
**Correctness:** 3
**Technical Novelty And Significance:** 3
**Empirical Novelty And Significance:** 3
**Recommendation:** 6

**Clarity, Quality, Novelty And Reproducibility:**

The paper is well-written, and critical information relevant to reproducibility is communicated extensively in the experiments section and appendix. Moreover, the authors situate their contributions well in the context of recent progress in understanding the role of geometry in SSL generalization. Further, I believe the loss-landscape analysis of contrastive/non-contrastive SSL is novel and presents a great starting point for designing augmentation strategies that lead to desirable representation characteristics.

**Strength And Weaknesses:**

Some strengths of the paper are outlined below:
+ The article is well-written and presents a coherent argument for disentangling the role of data augmentation in self-supervised learning via the lens of loss-landscape analysis.
+ The authors provide empirical evidence that suggests some qualitative alignment in the behavior of the SSL learning objectives on realistic models/datasets.

Some weaknesses of the current paper include the following:
- The analysis focuses almost entirely on linear models, with little discussion on the validity of the nonlinear regime.
- Though not essential, it would be helpful to have similar qualitative/quantitive experiments for more architectures (in particular transformers) to ablate the role of architecture design from a choice of the loss function.

**Summary Of The Paper:**

Self-Supervised Learning (SSL) algorithms like BarlowTwins and BYOL (both contrastive, non-contrastive) are known to suffer from $\textit{dimension collapse}$, where the effective rank of the representations is much smaller than the dimensionality of the representation. Empirical analysis of such algorithms presents conflicting evidence on the role of $\textit{dimensionality collapse}$ in downstream generalization. The authors give a different perspective, where they study the loss-landscape of SSL with simplified linear models. In particular, this analysis highlights the role of data augmentation & data-variation in inducing dimension collapse in the representations that could be beneficial for generalization by tuning out task-irrelevant dimensions. Building on these insights, the authors perform experiments w/ Resnet-18 on CIFAR datasets to demonstrate qualitative agreement with the theory.

**Summary Of The Review:**

In summary, the authors study the local geometry of the loss landscape with self-supervised learning objectives to understand the dimension-collapse in feature space. Using simplified analytic and linear models, the authors demonstrate that dimension collapse emerges by controlling the strength of data augmentation and data variation at train time and characterize conditions that lead to generalization-friendly dimension collapse. By extending experiments to non-linear ResNet-18 models on CIFAR datasets, the authors show qualitative similarities in loss landscape under different strengths of data augmentation.

---

> ### Author Response · Authors · 2022-11-15
> **author reply**
>
> Thank you so much for providing us with positive and constructive feedback! We are pleased to hear that you found our submission “well-written and presents a coherent argument for disentangling the role of data augmentation in SSL,” “the loss-landscape analysis … is novel and presents a great starting point for designing augmentation strategies that lead to desirable representation characteristics”. In the following, we would like to resolve all of your concerns.
>
> **"The analysis focuses almost entirely on linear models, with little discussion on the validity of the nonlinear regime."**
>
> Thank you for your question! We now dedicate the entire “Section 4.1” as well as “Appendix Section B” to thoroughly discuss and validate predictions of our theory in nonlinear models, which we have further expanded now with new experiments to reflect your feedback. Thorough empirical validation of our theory now includes
> - Landscapes of ResNet18 trained with varying strengths of data augmentation. (Fig. 2, top)
> - Landscapes of Vision Transformer trained with varying strengths of data augmentation. (Fig. 2, bottom)
> - ResNet models trained on datasets with varying degrees of imbalance, confirming our theoretical predictions on the Spectral Contrastive Loss over InfoNCE. (Fig. 4)
> - Validating “quantitative” predictions of our theory in a three-layer “nonlinear” network (App. Fig. 5)
> - Validating predictions of our theory in a three-layer nonlinear network “with normalization” (App. Fig. 6)
> - Thorough numerical comparison of landscapes between linear and nonlinear models, validating our results. (App. Fig. 7)
>
> We agree that theoretical analysis often requires a trade-off between interpretability and assumptions/simplifications we have to make. However, we’d like to emphasize that our current discussion successfully captures the most essential point: the loss functions are rotationally invariant, which in turn makes the odd-order terms in the Taylor expansion vanish, thus making the origin a special point in the landscape. Finally, we have empirically validated the robustness of our analysis in nonlinear settings by conducting experiments both on ResNet18 and Vision Transformer with non-linear activations (see below).
>
> **"Though not essential, it would be helpful to have similar qualitative/quantitive experiments for more architectures (in particular transformers) to ablate the role of architecture design from a choice of the loss function."**
>
> Thanks for your suggestion. We have newly performed the experiment and added the result for the vision transformer in Figure 2. The numerical results show that the qualitative picture is the same, which successfully ablates the role of architecture design from a choice of the loss function.
>
> **Summary**: Thank you again for carefully going through our analysis and making insightful suggestions to further robustify our claim. Following your suggestions, we now dedicate the entire Section 4.1 to discussing relevance to non-linear models and empirically validating our claim in realistic settings with both Resnet 18 and Vision Transformers. In light of the substantial updates, we now believe that your concerns are fully addressed, so we'd appreciate it if you could increase the rating and confidence score. Please let us know for any further clarifications.

---

### Author Response · Authors · 2022-11-15
**Summary of Change**

We thank the reviewers for their insightful feedback and unanimous support in accepting our manuscript. We are delighted to hear that the reviewers agree on the strong theoretical value of the theory: “novel and present a great starting point for designing augmentation strategies” [R ffca],” solid and comprehensive” [R m5m1], and “clear with a strong claim about the collapse in SSL” [R Q5CS].

To reflect the valuable feedback of the reviewers, we have made extensive updates and improved the manuscript (main updates are colored in orange):

1. We empirically study the loss landscape of Vision Transformers in Figure 2 and show that the main results still hold qualitatively

1. We add theoretical analysis of a non-gaussian, “structured” data augmentation that encourages scale invariance (Appendix F).

2. We ensure reproducibility by including a demonstration code that validates the main theoretical results of the paper

4. We add more intuitive discussions whenever possible to improve the readability to a broader audience

With these additions, we are confident that the updated manuscript is significantly improved. With the addition of the vision transformer example, our theoretical results are shown to hold for different large-scale architecture qualitatively. This broadens the scope of relevance of our theory. To the best of our knowledge, this is the first broadly relevant theoretical understanding of the collapse phenomenon from a landscape perspective. We believe that our theory will serve as a foundation for future theories of collapse.

Lastly, thanks again for your time, and please let us know if you have any further questions.

---

### Comment · Area_Chair_oWLB · 2022-11-26
**Following up on authors’ response and discussion**

Dear Reviewers,

Thank you very much again for performing this extremely valuable service to the ICLR community.

Please check the authors’ response and leave comments if you have not done it.

Best,

AC

---

### Decision · Program_Chairs · 2023-01-20

**Decision:**

Accept: poster

**Justification For Why Not Higher Score:**

This is a theory paper, primarily assuming linear models. Hence, the practical impact is not super.

**Justification For Why Not Lower Score:**

This is a solid theory paper, worth to share to the community.

**Metareview: Summary, Strengths And Weaknesses:**

The paper provides a theoretical explanation of the dimensional collapse problem in self-supervised learning. Unlike Jing et al. studies the learning dynamics of models for this purpose, this paper suggests the loss landscape around the origin as another way to understand this phenomenon. Specifically, the paper proposes an analytical theory on linear models and demonstrates that the proposed theory meets the empirical observations from non-linear models such as ResNet.

After the discussion phase, the reviewers unanimously supported the acceptance of this paper. The main concerns of the reviewers were the strong assumption of the linear model and Gaussian data augmentation and the limited numerical results. The author addressed these concerns by providing additional theoretical analysis on non-Gaussian data augmentation and additional experiments on non-linear models, including Vision Transformers.

**Note From Pc:**

if the above contains the word "oral" or "spotlight" please see: "oral" presentation means -> notable-top-5% and "spotlight" means -> notable-top-25%. As stated in our emails, we are disassociating presentation type from AC recommendations